# Polymer Modeling Reveals Interplay between Physical Properties of Chromosomal DNA and the Size and Distribution of Condensin-Based Chromatin Loops

**DOI:** 10.3390/genes14122193

**Published:** 2023-12-09

**Authors:** Daniel Kolbin, Benjamin L. Walker, Caitlin Hult, John Donoghue Stanton, David Adalsteinsson, M. Gregory Forest, Kerry Bloom

**Affiliations:** 1Department of Biology, University of North Carolina at Chapel Hill, Chapel Hill, NC 27599, USA; kolbin@live.unc.edu (D.K.); jstan73@live.unc.edu (J.D.S.); 2Department of Mathematics, University of California-Irvine, Irvine, CA 92697, USA; benjamlw@uci.edu; 3Department of Mathematics, Gettysburg College, Gettysburg, PA 17325, USA; 4Department of Mathematics and Carolina Center for Interdisciplinary Applied Mathematics, University of North Carolina at Chapel Hill, Chapel Hill, NC 27599, USA; david@unc.edu (D.A.); forest@unc.edu (M.G.F.); 5Applied Physical Sciences, University of North Carolina at Chapel Hill, Chapel Hill, NC 27599, USA

**Keywords:** chromatin organization, DNA loops, loop regulation, nuclear tethers, polymer modeling, crosslinkers, SMC complexes

## Abstract

Transient DNA loops occur throughout the genome due to thermal fluctuations of DNA and the function of SMC complex proteins such as condensin and cohesin. Transient crosslinking within and between chromosomes and loop extrusion by SMCs have profound effects on high-order chromatin organization and exhibit specificity in cell type, cell cycle stage, and cellular environment. SMC complexes anchor one end to DNA with the other extending some distance and retracting to form a loop. How cells regulate loop sizes and how loops distribute along chromatin are emerging questions. To understand loop size regulation, we employed bead–spring polymer chain models of chromatin and the activity of an SMC complex on chromatin. Our study shows that (1) the stiffness of the chromatin polymer chain, (2) the tensile stiffness of chromatin crosslinking complexes such as condensin, and (3) the strength of the internal or external tethering of chromatin chains cooperatively dictate the loop size distribution and compaction volume of induced chromatin domains. When strong DNA tethers are invoked, loop size distributions are tuned by condensin stiffness. When DNA tethers are released, loop size distributions are tuned by chromatin stiffness. In this three-way interaction, the presence and strength of tethering unexpectedly dictates chromatin conformation within a topological domain.

## 1. Introduction

Chromatin is dynamic, heterogeneous, and modified throughout the cell cycle. As a long chain polymer confined in a nuclear shell, chromatin is subject to random entropic forces (of order k_B_T), engagement with DNA and RNA polymerases, manipulation and bombardment by ATP-driven complexes (SMCs, topoisomerases, motor proteins, and helicases), and modification via DNA methylases/demethylases and chromatin-remodeling complexes. Collectively, these chromatin modifications influence its physical attributes, such as compaction and stiffness. Additionally, chromosomal loci can be physically constrained via tethering to external non-chromatin structures such as spindle microtubules, the nuclear lamina in interphase, matrix attachment regions (MARs), and scaffold attachment regions (SARs), or via internal tethering at the base of chromatin loops or by regions of entanglement [1,2,3,4,5]. Involved in a diversity of biological functions, tethering can be permanent and firm, as with the attachments at telomeres and centromeres in yeast [6,7,8,9,10,11]. Tethering can be transient and soft, as observed between promoters/enhancers at transcriptional hubs and during temporary relocation and association with transcriptional loci at nuclear periphery [12,13,14,15,16]. Tethering of a chromatin fiber at two ends can put DNA under enough tension to impact loop extrusion by resisting loop-extrusion forces [17,18]. Adding complexity to the extrusion process are the substrate properties of chromatin (such as stiffness). An easy-to-bend, floppy substrate versus a resistant-to-bending stiff substrate will conform differentially to loop extrusion forces.

Chromosomal loops were first observed by early microscopists in the late 1800s [19]. Loops as structural units of chromatin have emerged as central to the studies of chromatin organization and the regulation of gene expression [14,20,21,22,23]. At mesoscales, genomes are compartmentalized into regular looped subdomains that are topologically isolated and variable in size [23,24,25,26]. The mechanisms of stabilization and regulation of looped structures are under intense investigation [27,28,29]. Experimental observations support both loop extrusion and polymer phase separation as physical pathways behind the chromatin folding landscape [30,31,32]. Characterizing functional consequences of chromatin loops beyond their role in transcriptional regulation is a challenging proposal and is often aided by the use of modeling. One example of the functional importance of such organization is a centromeric bottlebrush, in which a regular array of loops at the centromeres of segregating chromosomes in yeast ensures critical tension sensing between sister kinetochores [33,34,35].

To understand loop regulation, we turned to bead–spring polymer chain models of chromatin and the activity of SMC proteins on chromatin. Polymer models of chromatin have proved critical to distilling the principles behind a number of processes including nucleolar organization [36,37], transcription [38,39,40], and DNA repair [41,42,43,44,45]. The investigation of high-level structure of looped chromatin compels consideration of its polymer features as a framework to establish the physical properties dictating chromatin’s response to small-scale forces [18,46,47]. Determining which forces prevail and how their balance is achieved requires dynamic modeling to dissect the contributions of individual components (chromatin, crosslinkers, and tethers).

DNA undergoes frequent transitions between heterochromatic compact fibers (condensed mitotic chromosomes) and less-compacted regions of chromosomal DNA (within topologically associated domains), or between nucleosome-occupied fibers and naked dsDNA/ssDNA molecules (transcription/replication) in growing cells. The stiffness of a polymer dictates the polymer’s response to entropic and enthalpic bending forces and is quantified by its persistence length (L_p_). Persistence length is the characteristic length scale over which the angular orientation of the two ends is correlated. Measurements of chromatin persistence length vary from approximately 200 nm, for highly compacted heterochromatic regions, to approximately 5 nm for ssDNA, and the average L_p_ of a naked double-stranded B-form DNA molecule is approximately 50 nm [48,49,50]. The presence of histones, scaffolding proteins, and chromatin modifiers on DNA allows for modulation of local persistence length in the range from 5 nm to 220 nm [50,51,52,53,54]. The resistance to bending of chromatin by its stiff form can result in far-ranging effects on chromatin loops, their sizes, and stability [55].

SMC complexes (cohesin, condensin, and Smc5/6) are essential to orchestrating higher-order chromatin organization. Post-translational modifications (PTMs) confer an astounding functional diversity onto SMC complexes. Residues on SMC and non-SMC subunits (e.g., kleisins) of both cohesin and condensin are also known to be heavily modified in response to emergent needs for chromatin reorganization and throughout the cell cycle [56,57,58,59]. Mutational scanning studies of condensin’s conserved ATP-binding sites point to distinct roles of subunits in chromosome morphology and genome architecture [60,61]. Expanding our understanding of DNA from being a passive substrate to having an active regulatory role, Varejão et al. demonstrated a physical interaction between a small region on Smc5p and a DNA fiber activating the ability of the Smc5/6 complex to modify proteins in its vicinity [62,63]. This is an example of direct feedback from chromatin to a crosslinker.

Studies depict the activity of SMC complexes as stochastic ratchets capable of stabilizing loops formed via Brownian motion, as in the case of cohesin [64] or condensin [65]. Others describe them as weak, directed mechanochemical motors, driving condensin [66] and cohesin [67] along the DNA fiber. Single-molecule in vitro experiments tracked condensins on DNA sheets, demonstrating their ability to unidirectionally translocate along DNA [66] and to extrude and stabilize loops in the presence of ATP [65,68]. Several models have described SMC complex loop-forming mechanisms in vivo, highlighting their generalized function as crosslinkers that form DNA-to-DNA bridges independent of ATP hydrolysis or the loop-extruding abilities of both condensin [69] or cohesin [70].

In the bead–spring polymer chain model of chromatin, we consider condensin as a simple Hookean spring crosslinking two chromatin beads. The condensin spring is stochastic and dynamic. First, it is seeded randomly on the bead–spring chain and is active for a duration representative of the lifetime of condensin on the DNA. Second, it can traverse the chain by stepping in a randomly chosen direction. It can crosslink distant parts of the chain in a single step, intermittently stall, or unbind. The condensin spring responds to chromatin tension and extends to its maximum length. Herein, we focus on a key parameter of the condensin crosslinker—the strength of the spring—as a coarse-grained depiction of post-transcriptionally modified SMCs.

Constraining the free movement of chromatin persistent and structural attachments (lamins, kinetochores, telomeres, Rabl-attachment sites, and the nucleolus) or transient and functional attachments (repair foci, PML bodies, and transcription factories) maintain the spatial chromosomal organization and chromosomal territories [71]. The Rabl-configuration constrains the motion of chromosomes proximal to attachment sites [8,72,73]. Chromatin tethers are likely to be nonrandomly distributed throughout the nucleus and vary widely in their resistance to motion [74]. Tethering can create sites of loop-rich and condensin-rich regions [17], which relax and mobilize in response to breaks in chromosomes [9,75]. Whether internal (*cis*- or *trans*- intrachromatin) or external (centromere–microtubule, telomere–nuclear envelope, or chromatin–nuclear lamins), nuclear tethers are diverse in nature. Tethers provide mechanisms to exert tension on the chromatin fiber. Davidson et al. showed that in the absence of tension on the DNA fiber, the boundary element CTCF failed to block cohesin extrusion, thus demonstrating feedback between chromatin tension and loop extrusion [76]. In simulations, we systematically varied tethering resistance to assess how tethering influences chromatin loops. Our modeling approach provides the opportunity to quantitatively evaluate the interplay between the tethering strength, chromatin stiffness, and condensin spring in regulating emergent loops of chromatin chains.

## 2. Materials and Methods

### 2.1. Computation Environment

The ChromoShake C++ code [34] was modified to run with ImageTank, a GUI simulation package from Visual Data Tools, Inc., Chapel Hill, NC, USA, which serves both as an interface for running simulations and a powerful data-processing software with a built-in 3D visualizer. Additional data visualization, processing, statistical analyses, and histograms and plots were rendered with DataGraph 5.1.2*beta* from Visual Data Tools, Inc., Chapel Hill, NC, USA.

### 2.2. Polymer Model of Chromatin

We employed a polymer bead–spring model of chromatin dynamics based on the model described in detail in our previous reports [18,34]. Chromatin was represented as a series of beads connected with linear springs. The beads were arranged in a linear chain connected via 100 simple Newtonian springs together at rest length representing a total of 1 μm of chromatin. A bead was interpreted as a single chromatin unit, and the amount of DNA it represented varies from ~30 bps (in the case of naked B-form DNA) to ~1000 bps (in the case of higher-order compact chromatin) [46]. Beads also possess an exclusion volume and are subject to Brownian motion [34] (see Appendix A). The dynamics were assumed to be non-inertial under the influence of high viscosity.

### 2.3. Persistence Length

To gain control over the persistence length of the simulated chromatin chain, we adjusted the strength of the hinge force that controlled the stiffness of the adjacent pair of spring segments in the chain with a hinge factor parameter. Beads in the adjacent spring segments were kept colinear by a restoring force, which was linearly scaled with the hinge factor (Appendix A). We empirically determined the hinge factors by running the model and finding values for which the radii of gyration of one-micron chains best matched the theoretically predicted radius of gyration, $R_{g}$ (Appendix A). The radius of gyration of a bead–spring polymer is defined and computed using Equation (1), where $r_{i}$ is the position of a monomer (bead), and $r_{mean}$ is the center of mass of all monomers (beads):(1)$${\langle R_{g}\rangle }^{2}=\frac{1}{n}\sum_{k=1}^{n}{\left|r_{i}-r_{mean}\right|}^{2}$$

When an entropic worm-like linear chain collapses into a random coil, the expected value of the chain’s gyration radius $\langle R_{g}\rangle$ and the average of its end-to-end distance $\langle R\rangle$ are related via Equations (2) and (3), respectively, where *N* is the number of Kuhn units, and *b* is the length of one Kuhn segment, equivalent to twice the persistence length L_p_:(2)$${\langle R_{g}\rangle }^{2}=\frac{N}{6}b^{2}$$
(3)$${\langle R\rangle }^{2}=Nb^{2}$$

Appendix A captures the empirically established relationship between the hinge factor parameter and the plateau values of the radii of gyration for the simulated chains in the model. The three hinge factors used in our experiments were picked as best approximations to match the theoretically expected $R_{g}$ and are listed in Table 1. The theoretical $R_{g}$ minimum for a linear one-micron chain of L_p_ of 50 nm with loose ends is 129.10 nm.

### 2.4. Tethering Resistance

Tethering in the model restricts the movement of the tethered beads and, therefore, influences the movement of the entire chain. The resistance to movement is via higher viscous drag achieved by proportionally scaling down the forces acting on the tethered (end) beads. Equation (4) captures how increasing the tethering resistance (AU, unitless) decreases the force acting on a tethered bead, in effect, reducing its motion:(4)$$F_{on\;tethered\;bead}=\frac{1}{Tethering\;resistance\;(AU)}F_{on\;bead}$$

Using Equation (1), we computed the radius of gyration for the L_p_ = 50 nm linear chain for a range of values of tethering resistance. When the linear chain is extremely stiff, it behaves like a rod, and its $R_{g}$ reaches a maximum. When a one-micron chain with anchored (immobile) ends is extended to its full contour length (1 μm), its maximum $R_{g}$ is computed as 288.68 nm (the empirically calculated value). Over a finite time of simulation, equivalent to approximately 8 min of biological time (for the time conversion see Table 2), dropping the tethering resistance (from 10^7^ to 1 AU) allows the chain to explore more states due to thermal motion, such that respective values of $R_{g}$ at their plateaus drop accordingly for each regime, and we see the range of average $\langle R_{g}\rangle$ values from the maximum to the minimum (Appendix A). For a linear one-micron chain of L_p_ = 50 nm, the values ranged from 288.68 nm to 129.10 nm (Appendix A and Table 1).

### 2.5. Condensin Springs

Condensins were modeled as dynamic springs crosslinking pairs of beads. A condensin spring takes a step by releasing from one bead. A vector centered at the released bead extends 10 nm away from the bound bead and searches for a new bead. Crosslinking to the new bead completes one step of condensin extrusion. The period (τ) between crosslinking steps is exponentially distributed with mean $\tau_{0}$ and is obtained using Equation (5):(5)$$\tau =\tau_{0}{\left(\frac{l_{t}}{l_{0}}\right)}^{},$$
where $l_{t}$ is the length of the maximally extended DNA spring immediately adjacent to the condensin crosslink, representing local tension on the chromatin, and $l_{0}$ is the DNA spring under no tension, representing its rest length [18]. As the local chromatin tension is increased, the stepping is delayed proportional to tension. This throttling of condensin in response to high chromatin tension can lead to occasional stalling, a biological feature of condensin demonstrated in previous reports [68]. As in our previous report, the condensin spring Young’s modulus was set to 2.0 GPa [17], and we changed it over three orders of magnitude, reflecting the potential biological consequences of PTMs, to obtain strong, moderate, and weak springs.

Condensin springs can extend up to a critical length of 30 nm [17]. When a condensin spring exceeds its critical length, it must unbind one bead and search the vicinity for the nearest bead. The stepping of the leading bead coupled with the release of the lagging bead can produce directional walking along the chain when chains are linearly extended (Appendix A). On linear chains, the average condensin step rate is approximately two beads per second. If the linear chain represents naked B-form DNA (~30 bps per bead), the step rate equals 60 bps/s, as found in in vitro studies [66] and our previous model [17].

The location of the nearest bead depends on the chain’s topology at the time, such that on randomly coiled chains, crosslinking the nearest bead results in *trans* (distant beads) as well as *cis* (adjacent beads) connections. The ability of condensin to crosslink distant parts of a chain after only taking a single step (called “diffusion capture”) has been demonstrated by others with simulations and in vitro, and it permits condensins to engage in both *cis*–*cis* and *cis*–*trans* crosslinks [79]. Condensins can unbind both beads randomly at the average rate of 0.01135 events per second of simulation time. Less frequently, condensins reverse the stepping direction by randomly swapping their leading and lagging beads at a rate of approximately 0.02 events per second of simulation time [66,68].

In this report, we changed the strengths of condensin springs to simulate the putative mechanisms of condensin regulation in the cell. Table 3 summarizes the relevant features of the modeled condensin springs based on earlier simulations and supported by biophysical reports from other groups.

### 2.6. Recoil Imaging

Dicentric strains KBY6201a (Spc29-RFP) and KBY8182 (Spc29-RFP, *brn1-9*) were imaged for the measurement of recoil dynamics (Appendix A) and are listed in Table 4. Cells were grown in YPG to reach logarithmic growth at 24 °C and then shifted to YPD (dicentric activation) and kept for 3–6 h at either 24 °C or 32 °C to induce temperature-sensitive phenotypes for the *brn1-9* strain. Timelapses of early anaphase cells were acquired as described previously [80]. Recoil events and cell percentages with snapback events were visualized in the KBY7002 (Tub1-GFP) dicentric strain (Appendix A). Cells were imaged at room temperature (24 °C) using a Nikon TE2000-U widefield microscope with a 60× Plan Apo 1.40 NA Nikon objective and a Hamamatsu Orca-Flash4.0 LT camera using MetaMorph 7.8.10.0 imaging software (Molecular Devices, LLC, San Jose, CA, USA). Every 30 s for the duration of the timelapse (15 min), a 5-step Z-stack with a 300 nm step size was taken in the GFP (300 ms exposure) and RFP (300 ms exposure) channels with a single transillumination (100 ms exposure) in the home plane.

### 2.7. Image Analysis

Each Z-stack was compiled into a max. intensity projection using a custom MetaMorph journal, and SPB-SPB distances in the KBY6201a and KBY8128 strains were measured from the brightest pixel of the first SPB to the brightest pixel of the second SPB using FIJI (ImageJ 2.9.0, NIH, Bethesda, MD, USA). The first measurement was made for a frame in which a recoil event was evident, and measurements continued for all subsequent frames collected. The means and standard deviations were calculated and displayed in violin plots using DataGraph 5.1.2*beta*, Chapel Hill, NC, USA. Kymographs were constructed by drawing a line (with a width of 20 pixels) coincident with the spindle, followed by using the Kymograph Builder FIJI plugin, Release 1.2.4 [81]. Each pixel on the kymograph’s *x*-axis represents one frame of 30 s.

## 3. Results

### 3.1. Condensin Spring Strength Dictates Distribution of Loop Sizes on Chains

Three examples from single instantiations (Figure 1A) show the average DNA loops formed by condensin springs of varying strengths. The global frequency distribution of all loop sizes as a function of the condensin spring strength and the persistence length (stiffness) of the DNA chains is summarized in a histogram (Figure 1B). The distribution of loops (*y*-axis) formed on chains whose ends are anchored depends only on the strength of the condensin spring and is not influenced by chromatin stiffness (i.e., chain L_p_, *x*-axis).

The largest loops of approximately 70 beads are formed by strong condensins with an average of ~23.5 beads/loop (Figure 1A,B, top panels). The maximum loop size for moderate condensins is approximately 20 beads with an average of ~8.3 beads/loop (Figure 1A,B, middle panels). The smallest loops are formed by weak condensins, with the maximum size reaching approximately 5 beads/loop and an average of ~2.1 beads/loop (Figure 1A,B, bottom panels). On anchored chains, only strong condensin springs can form large loops. The largest loops (~70 beads/loop) constitute approximately two-thirds of the total chain length. The remaining one-third of the chain (i.e., ~30 bead–springs outside of a loop) covers the entire 1 μm distance between anchors, with individual springs extending up to 30 nm, which is three times their original rest length (10 nm). On anchored chains, moderate condensin springs can generate loops reaching a maximum of approximately ~20 beads/loop.

### 3.2. Kinetics of Chromatin Compaction in Live Cells

To compare the model predictions of condensin function to chromatin compaction in vivo, we utilized the mitotic spindle to apply a force to a single chromosome in a live cell. We introduced a conditionally functional dicentric chromosome into yeast. The conditional centromere is activated upon growth in glucose. When the centromeres on the same sister chromatid attach to opposite spindle poles, the chromosome becomes stretched between the two poles, resulting in a cell cycle pause in the mid-anaphase [82]. Upon spindle breakdown, the recoil of the dicentric chromosome draws the spindle poles together as the microtubule-based spindle collapses. The rate and extent of recoil provide a kinetic assay for condensin-based chromatin compaction in live cells. We labeled the spindle pole protein (Spc29) with a red fluorescent protein (RFP) to track the motions of SPBs during their recoil. The rates of recoil were equivalent in the wild type and *brn1-9* (condensin subunit) mutants. Spindle collapse was complete after approximately 3 min, at which time the spindle poles remained separated by 1–2 µm (Figure 2D). The SPB-SPB distances in cells with the temperature-sensitive allele of condensin (*brn1-9* at 24 °C and 32 °C) were at least two-fold greater than in cells with functional condensin (WT). The DNA between the poles collapsed to a final volume measured as the distance between SPBs of 0.69 μm in cells with condensin and 1.53 μm and 1.83 μm in cells with moderate and severe condensin reduction, respectively (Figure 2D).

To examine the expected contribution of condensin to in vivo kinetics and the degree of compaction, we examined the simulations of DNA recoil following the release of one end of a tethered DNA molecule (Figure 2E). The end-to-end distance decreased at slower rates with fewer condensins on the polymer chains. Plateaus of approximately 40 nm, 120 nm, and 320 nm were reached for six, three, and one simulated condensin, respectively (Figure 2E). A single condensin was sufficient to reduce the end-to-end distance of a linear chain (starting configuration) by over half early in the simulation (t < 5 ms). Reducing the number of condensin springs on the DNA in simulations qualitatively recapitulated the compaction (Figure 2F) observed in the in vivo measurements in *brn1-9* mutants, which lacked functional condensins compared with the wild-type cells.

### 3.3. Stiffness of the DNA Chain Dictates Distribution of Loop Sizes on Unconstrained Chains

The ability of condensin to extrude loops and compact the chain is dependent on the stiffness and tension of the chromatin chain [18]. Regions of high chromatin tension resist the extrusion activity of condensin and could force condensin into a stalled state. Unconstrained chains quickly conformed to random coils and occupied a space proportional to their persistence lengths (Figure 3B, Table 1). On unconstrained chains, the persistence length (L_p_) (Figure 3A, *x*-axis) emerged as dominant in dictating loop sizes. The tensile strength of condensin springs (Figure 3A, *y*-axis) had a negligible effect on the loop size distribution.

Overall, the distributions shifted toward smaller loops across all regimes, and stiffer chains resisted the formation of larger loops. Large loops could be formed by condensin springs of each strength on the chains of all three L_p_ values (see the distribution of tails on the right), but they were much less frequent than smaller loops. On anchored chains, large loops were only formed by strong condensin springs (Figure 1B). Conversely, large loops were quite rare on chains with no constraints on the ends; however, they were present in all parameter combinations.

### 3.4. Persistence Length and Condensin Spring Strength Both Determine Loop Sizes in Moderate Tethering Regimes

Chromatin tethers represent a hindrance to the motion of the chain. For example, in strong confinement, at centromeres and telomeres, the tethers are well-defined as microtubule- or nuclear envelope-binding sites. However, entanglement regions or interactions with protein scaffolds present as “soft” or transient tethers and also restrict chain motion. We therefore introduced variable tethering by allowing a degree of constraint on each end. We considered a range of tethering resistance from 10^7^ AU to 10^2^ AU for a one-micron chain while quantifying loop sizes as above and classified tethers as hard or soft depending on an arbitrary unit value corresponding to their resistance to motion. At the high end of tethering resistance, the chain behaved as if it was anchored. As the tethering resistance on the ends decreased, the chain mobility increased, and, in the presence of thermal noise, the chain more readily collapsed into a random coil (Appendix A). The addition of the dynamic crosslinking of the linear chain by condensin springs further facilitated chain collapse as each bead–bead crosslinking event propagated throughout the whole chain (Appendix A, with tethering at 10^3^–10^5^ AU, respectively; single instantiations with one condensin are shown).

Decreasing the tethering resistance by one order of magnitude from 10^7^ to 10^6^ changed the size distribution of loops formed only by the strong condensin spring (Appendix A). At a tethering resistance of 10^6^ AU, the stiffness of the chain emerged as a relevant parameter determining the sizes of loops formed by the strong condensin (Figure 4, top panels). As the tethering resistance was reduced to 10^5^ AU, the sizes of loops formed by the moderate condensin spring began to differentiate according to the chromatin stiffness (Figure 4, middle panels). Loops formed by the weak condensin spring increased on average when the tethering resistance was reduced from 10^6^ AU to 10^5^ AU. However, within the same tethering regime (10^5^ AU), loops formed by the weak condensin spring were not differentiated by size according to the chromatin stiffness (Figure 4, bottom panels).

In the softer tethering regimes (Figure 4, tethering resistance of 10^4^ AU), the size distributions of the loops formed by the strong, moderate, and weak condensin springs look strikingly similar to each other and are all differentially affected by the chromatin persistence length. At lower values (less than 10^4^ AU) of tethering resistance, the loop size distribution histograms suggest that chromatin stiffness dominates the strength of the condensin crosslinker and emerges as the dominant physical parameter determining loop sizes (Appendix A).

The distributions of loops displayed in the histograms of Figure 3A (unconstrained ends) and Figure 1B (anchored ends) indicate that the combined influence of physical properties, such as the tensile strength of condensin (Figure 1), the number of condensins (Figure 2) and substrate stiffness (Figure 3), together with the presence or absence of tethers on chains, are biologically active tuning parameters in determining loop sizes and distributions throughout the genome.

### 3.5. Persistence Length of DNA within a Loop Determines the Space It Will Explore

Loop size in the model is measured in beads and provides information regarding the relative amount of DNA in the loop. However, loop size is not the only parameter influencing how the loop explores space or the volume it occupies. The persistence length of DNA is a crucial parameter required to estimate the space explored by the loop. We calculate the radii of gyration as statistical measures of space occupied by same-size loops of varying stiffness. A strong condensin spring could form same-size loops (66 beads) on chains of three persistence lengths (L_p_ = 5 nm, 50 nm, and 150 nm) in our model, each with significantly different radii of gyration (Figure 5A). An equivalent amount of DNA in a very stiff loop explored on average two-fold more space compared with a floppy loop (Figure 5A, see inset in top panels). The observation of stiff loops exploring more space was also seen for smaller loops (20 beads) of different stiffnesses formed by moderate condensin springs (Figure 5B, bottom panels). Thus, the volume explored by the loop increased with increasing chain stiffness. This result was predominantly the consequence of a polymer responding to entropic forces (k_B_T).

The volume explored is dependent on the persistence length of the polymer and is independent of the crosslinker strength (Figure 5B). The stiffness of chromatin can influence the space the DNA loop explores and should be considered as a parameter with the potential to affect the biological functions of the domains in the loop.

## 4. Discussion

The ability of cells to access genetic information relies, in part, on the spatial organization of chromosomes within the nucleus. Major strides in understanding higher-order structures, beyond nucleosomes, have come from the discovery of the loop-extruding and crosslinking properties of the class of proteins known as structural maintenance of chromosomes Smc1/3 (cohesin), Smc2/4 (condensin), and Smc5/6. Equally impactful is the realization that the physical properties of their substrate, chromatin, contribute to the structural organization. The substrate properties are likely tuned by solvent variations in the nucleoplasm via cellular mechanisms open for further study. Using modeling and biological validation, herein, we demonstrated how the stiffness of both the chromatin fiber and the condensin complex, as well as chromatin tethers, impact the size and distribution of intra-chromosomal loops.

A bead–spring polymer framework provides the platform for exploring mechanisms from the microscale of protein function to the macroscale of chromosome dynamics. A bead–spring polymer is constantly fluctuating in such a way that intra-molecular loops are constantly being generated and dissolved. In cells, these loops are a means to sequester biochemical activities through the topological control of the loop and/or the spatial proximity of the looped sequences. The distribution of loops in vivo, unlike in the thermal model, is not random but rather exhibits patterns specific to the cell type, cell cycle, and locus.

The major drivers of loop size and distribution are the strength and the number of crosslinking proteins such as condensin, the stiffness of the chromatin, and the strength of tethers that anchor the regions of chromatin. How these parameters interact and feedback to one another is not completely intuitive. It is likely that all these parameters are subject to cellular regulation, and together contribute to the complex pattern of loop heterogeneity observed in vivo. While this study focused on the distribution and number of loops on a single chain, similar dependencies would, in principle, contribute to loop formation in an environment with multiple chains and crowded with cellular components. It is, therefore, imperative to build physical representations of these features to be able to dissect the targets of cellular regulation.

Condensin tensile stiffness is the dominant driver of loop sizes in chromatin with strong or immovable end tethers (Figure 1). In this regime, the DNA chain is firmly anchored at the ends, such that the chain is under tension. Firm anchors are defined as those that confer tension to the DNA and persist over timescales of biological significance (i.e., minutes). The most prominent is the example of centromere binding via the kinetochore to spindle microtubules. Firm anchors include the tethering of telomeres and active genes to the nuclear envelope. The contribution from chain DNA stiffness to loop formation is dwarfed by the tension applied by anchoring the ends and restricting the degrees of freedom of the chain.

For regions of the genome that are not tethered, and thus exhibit a greater range of motion, the stiffness of the chain is the major driver of loop size and distribution (Figure 3). In this situation, floppy chains result in beads separated by long linear distances (i.e., base pairs) coming into proximity, giving rise to longer and broader distributions of DNA loops. The finding that chain stiffness drives loop size distribution was observed in the relative insensitivity to condensin spring strength (Figure 3). In vivo, the situation is likely to lie between the two extremes of firmly tethered and non-tethered. The relationship between chain stiffness and spring strength in cells in a range of tethering regimes reveals numerous ways to attain similar distributions. The redundancy in the system contributes to its robustness as well as specificity. The regulation of tethering strength may be broadly invoked by the cell, in addition to its proposed role in the repair of chromosomal breaks [9,75].

The role of crosslinkers and the distribution of loops have impacts well beyond gene expression. Regions of very high loop density were originally shown to function as a tension-producing bottlebrush. The loop density provides a way to stiffen the primary axis at the base of loops and thus oppose forces generated at the spindle. The bottlebrush is an essential component in the force balance between the chromosome and spindle microtubules required for high-fidelity chromosome segregation. In the nucleolus, the loops facilitate the ability of RNA polymerase to transcribe rDNA and meet the protein translation needs for cellular life. In addition, crosslinkers at the base of loops can associate, revealing how neighborhoods within the genome are assembled. The dynamics of the crosslinkers have the remarkable ability to regulate the homo- or heterogeneity of the rDNA genes via slow (homogeneous) or fast (clustered) crosslinking [18,83]. In this case, weak, local crosslinking within segments of a large-molecular-weight chromosome induces global emergent behavior, either amorphous and miscible or compact with frequent self-interactions.

The multiple modes of chromatin modification together with crosslinkers and loop extrusion capability result in a staggering degree of variation in loop size and distribution. The analysis of how the variation scales with different physical properties of the chromatin and condensin serves to demystify the heterogeneity and dynamics observed in live cells.

## Figures and Tables

**Figure 1 genes-14-02193-f001:**
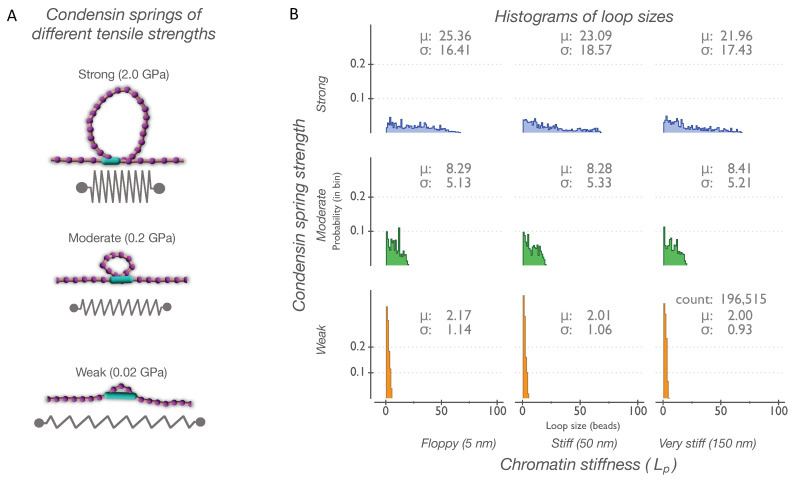
Condensin spring strength dictates distribution of loop sizes on DNA chains with anchored ends. (**A**) Example visualization of loops formed by condensin springs of different tensile strengths on a stiff chain with anchored ends. Loops are measured in beads, and an average loop for each condensin spring is shown (strong, 23 beads; moderate, 9 beads; and weak, 3 beads). Chromatin bead–spring chain is purple, and condensin spring is shown as a light blue cylinder. (**B**) Histograms of loop sizes formed by condensin springs (weak; moderate; and strong) as a function of persistence length (L_p_ = 5 nm; L_p_ = 50 nm; and L_p_ = 150 nm). Discrete loops measured in beads are binned by size ranging from 1 to 99 (bin size = 1). Heights of bins indicate probability of a loop occurring (probability density function) in that bin. Data for each histogram tile were obtained from randomly seeded independent runs (*N* = 20), and a minimum of 196,515 counts were collected from 35 ms of simulation.

**Figure 2 genes-14-02193-f002:**
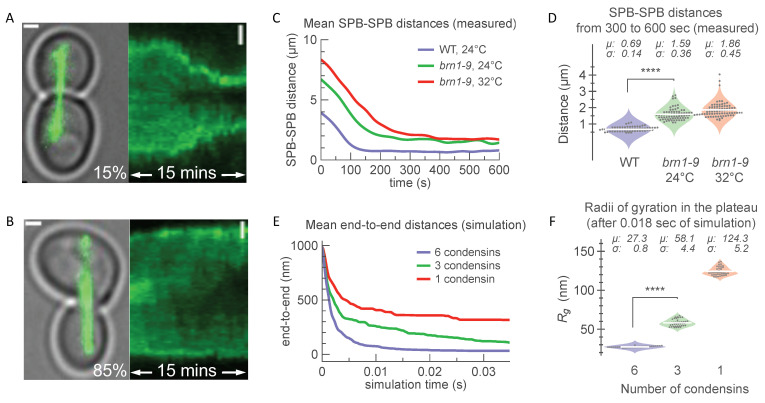
In vivo single-molecule experiment of spindle collapse via DNA recoil compared with collapse of one-micron DNA chain with simulated condensins. (**A**) Characteristic transillumination image of yeast cells in late-anaphase prior to SPB-SPB snapback (15% of events). Tubulin-GFP is shown, and spindle poles are dense signal clusters at distal ends. On the right, kymograph of GFP signal in mother and daughter cells is shown, where SPBs recoil from their original positions (on left, t = 0 min) toward the bud neck (on right, t = 15 min). (**B**) Same as above, showing t = 0 min cell snapshot and kymograph (0 ≤ t ≤ 15, in mins) in the absence of SPB-SPB snapback (85% of events). Scale bar for all images is 1 µm. (**C**) Mean distances between spindle pole bodies (SPBs) in snapback events measured over 10 min in wild-type dicentric (*purple*, *N* = 12) and *brn1-9* dicentric strains at 24 °C (*green*, *N* = 10) and 32 °C (*red*, *N* = 14). (**D**) Violin plots with mean SPB-SPB distances post-snapback (i.e., average of 300 ≤ t ≤ 600, in seconds) for wild type at 24 °C (*purple*); *brn1-9* mutants at 24 °C (permissive temperature, *green*; and *brn1-9* mutants at 32 °C (sensitive temperature, *red*). Individual measurements collected every 30 s are grouped by condition (grey dots). Average distances and means are provided above. Comparison between WT and *brn1-9* mutant dicentrics at 24 °C is shown significant by a single-factor ANOVA test (summarized in Appendix A) as indicated by **** (*p*-value < 0.001). (**E**,**F**) Simulations of condensin springs on one-micron chain. (**E**) Reduction in condensin number (from 6 to 1) increases the end-to-end distance trajectories and the rate of chain collapse, and (**F**) the final volume (*R_g_*, radii of gyration) of the chain. Simulation time of 35 ms equals to approximately 8 min of microscopic measurement time (see Table 2). Data for trajectories in (**E**) are averages of independent randomly seeded simulations (*N* = 20) with 6 condensins (*purple*), 3 condensins (*green*), and 1 condensin (*red*) on L_p_ = 50 nm chains whose ends are tethered, providing a drag with resistance value of 10^4^ AU (see example of single instantiation with one condensin spring in Appendix A). Data for (**F**) are values from (*N* = 42) observations taken at regular time intervals within the plateau regions for each group (0.018 ≤ time ≤ 0.032, simulation seconds). Comparison between 6 and 3 simulated condensins is shown significant by a single-factor ANOVA test (summarized in Appendix A) as indicated by **** (*p*-value < 0.001).

**Figure 3 genes-14-02193-f003:**
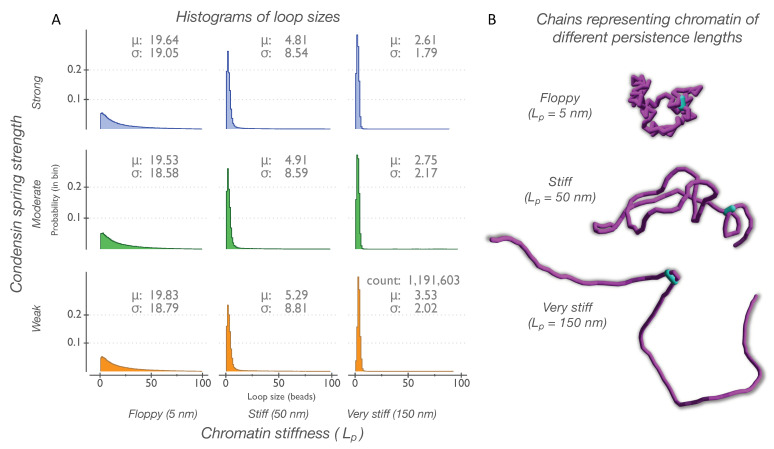
Persistence length of DNA chain dictates distribution of loop sizes on chains with unconstrained ends. (**A**) Histograms of loop sizes formed by condensin springs (weak; moderate; and strong) as a function of persistence length (L_p_ = 5 nm; L_p_ = 50 nm; and L_p_ = 150 nm). Discrete loops measured in beads are binned by size ranging from 1 to 99 (bin size = 1). Heights of bins indicate probability of a loop occurring in that bin (probability density function). Data for each histogram tile were obtained from randomly seeded independent runs (*N* = 20), and a minimum of 1,000,000 counts were collected from at least 35 ms of simulation. (**B**) Filled volume representations of chromatin bead–spring chains (purple) are shown. Condensin springs are shown as light blue cylinders.

**Figure 4 genes-14-02193-f004:**
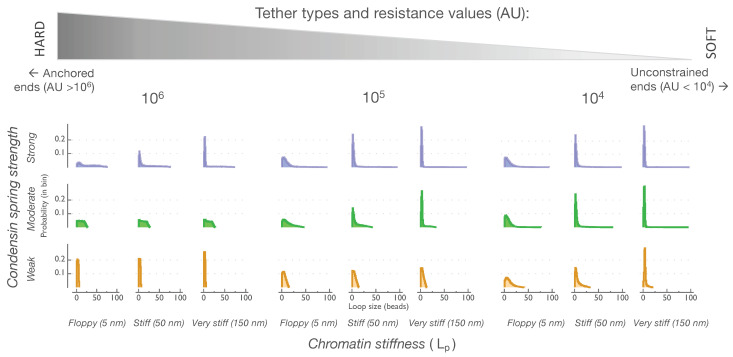
Persistence length and condensin spring strength *both* determine loop sizes in moderate tethering regimes. Loops are quantified in histograms as described above (Figure 1 and Figure 3). For resistance values of 10^6^ AU to 10^4^ AU, the strength of tethering reveals regimes in which individual condensin spring responds differentially to change in chromatin stiffness. Strong condensin spring (**top**, on left) responds differentially to chromatin stiffness at tethering resistance of 10^6^ AU and below. Moderate spring (**middle**, at center) responds differentially at tethering resistance of 10^5^ AU and below. Weak spring (**right**, on bottom) responds differentially at tethering resistance of 10^4^ AU. Histograms of loop size distributions for tethering resistance above 10^6^ AU and below 10^4^ AU are provided in Appendix A. As tethers soften (**left**–**right**), the resistance on end beads is reduced by a factor of 10, and regulation of loop sizes transitions from control by condensins (compare with Figure 1) to control by chromatin stiffness (compare with Figure 3).

**Figure 5 genes-14-02193-f005:**
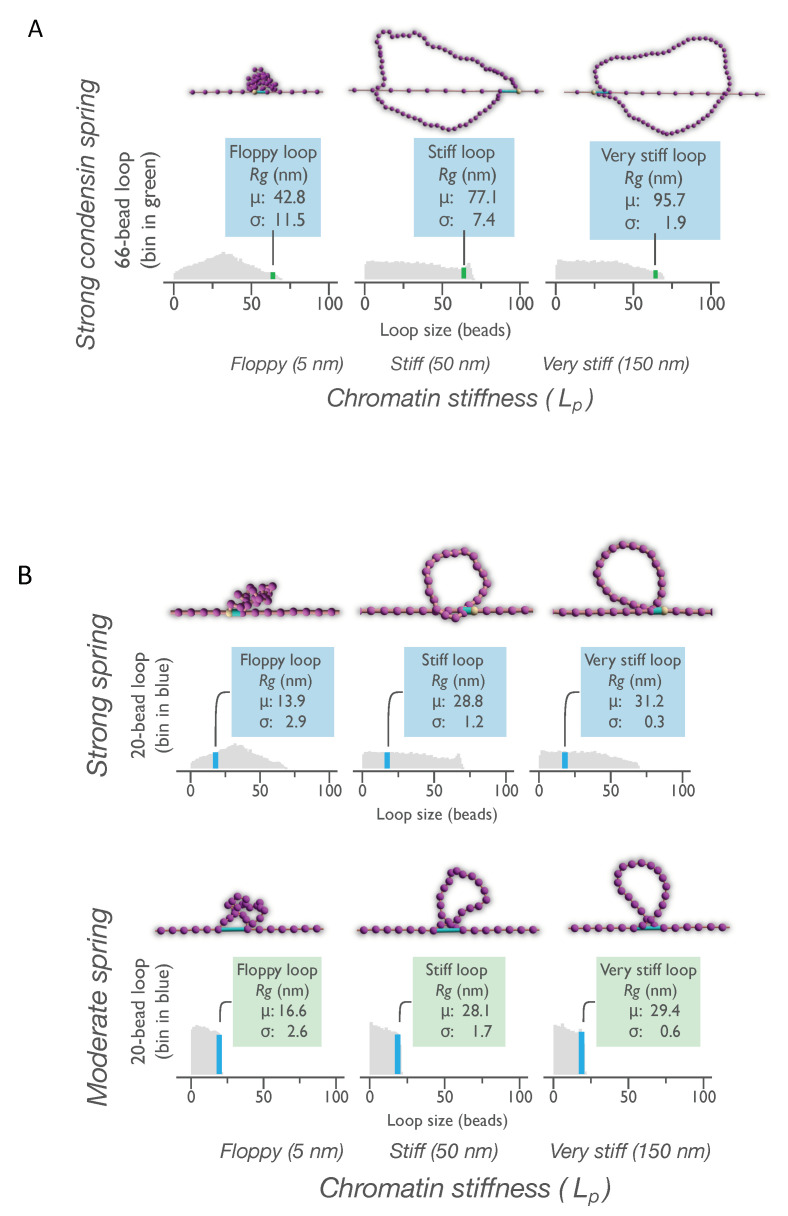
Persistence length of DNA within a loop determines its volume (*R_g_*), and loop volume does not depend on condensin spring strength. (**A**) Radii of gyration (*R_g,_* nm) occupied by beads within large chromatin loops of varying stiffness (66-bead loops, green bins) formed by strong condensin). Differences in loop *R_g_* between groups are significant with *p*-value approaching 0. (**B**) Radii of gyration (nm) occupied by smaller loops of varying stiffness (20-bead loops, blue bins) formed by strong or moderate condensin. In both (**A**,**B**), arrows show bins containing loops of indicated size from distributions obtained on one-micron chains with hard tethers (tethering resistance of 10^7^ AU, Appendix A). Chromatin bead–spring chain is purple, and condensin spring is shown as a light blue cylinder. Loops formed by weak condensin springs on anchored (Figure 1) or hard-tethered (Appendix A) chains are too small (max ~5 beads) to compute volume. The means and SDs (above) were calculated by averaging *R_g_*s at each instantiation of indicated size loop in the bin collected from at least 35 ms of simulation. For summary statistics, see Appendix A.

**Table 1 genes-14-02193-t001:** Correspondence between persistence length and modeled parameter, hinge factor, for a linear one-micron chain. Expected *Rg*s were calculated using formula for mean *Rg* of ideal linear chains (*Nb*^2^/6)^1/2^ from the study by Rubinstein and Colby [77] (pg. 60–63).

Persistence Length, L_p_ (nm)	Kuhn’s Length, *b* (nm)	Segments, *N*	$\mathrm{Expected}\;R_{g}$ (nm)	Measured Mean $R_{g}$ (nm) *	Hinge Factor (Unitless)
5	10	100	40.82	39.97	0
50	100	10	129.10	130.68	3.25
150	300	3.333	223.61	221.80	16

* means of *N* = 8 independent simulations.

**Table 2 genes-14-02193-t002:** Time conversion.

Time	Value
Time conversion factor **	14,100×
Maximum computed simulation time	35 ms
Converted simulation time (14,100 × 0.035 s)	~8 min
Time-lapse experiment duration (max. 600 s)	6–10 min

** Based on viscosity in the study by Fisher et al. [78].

**Table 3 genes-14-02193-t003:** Condensin spring features.

Feature ^1^	Value	Reference
Translocation rate (due to directional stepping/extrusion)	60 bps/s	[17,66]
Throttling/stalling/tension sensing	N/A	[18,68]
Unbinding and rebinding rates, and reversal rate	0.01135 events/s; 0.02 events/s	[68]
Diffusion capture	N/A	[18,79]
Condensin spring strength and flexibility	2.0 GPa; 0.2 GPa; 0.02 GPa	[17], This study
Condensin spring critical length	30 nm	[17,18]

^1^ See Appendix A for select visualizations of condensin springs.

**Table 4 genes-14-02193-t004:** Yeast strains summary.

Described as	Strain	Genotype
Tubulin-GFP (dicentric)	KBY7004	KBY4137/J178D dicentric URA3::Hg + TUB1:GFP:URA3 pAFS125
WT (dicentric)	KBY6201a	*ade2-1*; *can1-100*; *his3-11,15*::GFP-LacI-HIS3; *leu2-3,112*::lacO-LEU2; *trp1-1*; *ura3-1*::URA3GALCEN3; Spc29RFP:Hb
*brn1-9* (dicentric)	KBY8182	*ade2-1*; *can1-100*; *his3-11,15*::GFP-LacI-HIS3; *leu2-3,112*::lacO-LEU2; *trp1-1*; *ura3-1*::URA3GALCEN3; Spc29RFP:Hb; *brn1-9*:Nat

## Data Availability

The simulation code and all analysis scripts are written in C++ and were run using ImageTank as the user interface. All simulation scripts and some data analysis scripts have been made available as .itank files at https://github.com/kolbincode/ChromoShake2023 (accessed on 28 August 2023). The data used for figures, statistical summaries, and analyses are contained in the Appendix A. The software used were DataGraph (Software Version 5.1.2*beta*, Visual Data Tools, Inc., Chapel Hill, NC, USA, https://www.visualdatatools.com, accessed on 8 August 2023) and ImageTank (Visual Data Tools, Inc., Chapel Hill, NC, USA, https://www.visualdatatools.com, software downloaded and accessed on 8 August 2023).

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
