# Peer review of "Polymer Modeling Reveals Interplay between Physical Properties of Chromosomal DNA and the Size and Distribution of Condensin-Based Chromatin Loops"

_genes, 2023, doi:10.3390/genes14122193_

Round 1
Reviewer 1 Report
Comments and Suggestions for Authors
In this paper, Kolbin et al. investigate by polymer modeling the link between physical properties of chromatin (chromatin stiffness, stiffness of crosslinkers such as cohesin, strength of chromatin tethers such as centromere to the microtubule) and the formation of chromatin loops, highlighting that different properties dominate in different conditions in determining the number of loops formed, their size and volume distributions. I find the manuscript of interest and well-written, and I only have a couple of (minor) concerns:
- The authors stress that the distributions of size and number of loops largely varies based on several factors (as well as that different combination of those factors can drive similar distributions). In this regard, it would be convenient to compare (or at least add a comment about the matter) the results with the experimental distribution of loops found in the literature to clarify the distribution in the model that is more realistic and on the sets of paraments that allow obtaining the best agreement with available data (e.g., obtained by analyzing loops from available Hi-C data).
- The authors correctly consider loop extrusion as an organizing mechanism of chromatin structure; however, additional mechanisms have been shown to play a role as well, notably phase separation (see, e.g., a recent review in Int. J. Mol. Sci. 2023, 24(4), 3660). In this regard, it would be convenient to comment on how this essential element could be included and affect the model, e.g., in the Introduction or Discussion section.
Author Response
To Reviewer,
Thank you for your careful consideration of our work. Please, see the attachment for point-by-point response to your suggestions.

Reviewer 2 Report
Comments and Suggestions for Authors
This is a very interesting paper. The authors wrote a very comprehnsive and insightful introduction to the subject of loop extrusion. The authors produced insightful novel data measuring with precision the recoil of the two spindle pole bodies on a dicentric chromosome in mid-anaphase in baker's yeast, in both Wild Type as well as condensin mutants. This is a beautiful and to my knowledge novel approach, that have the potential of providing great details about the chromatin mechanics in this phase of the cell cycle. I wish that the authors will have the opportunity to pursue and deepen this research.
They decided to approach this dataset through a boastful modeling-first-strategy, using a very detailed coarse grained molecular dynamics simulation that in my opinon misses most of the landmarks of the experimental dataset they produced (figure 2). The modeling also provides many predictions for which every reader will have the right to reserve their doubts about their validity as I do as a reviewer.
The discussion is very interesting.
I reccomend the publication of the results after minor corrections.
Minor comments:
- section 2.2: Please include the equation of motion for the polymer, or the mathamatical description of the energy terms governing the motion.
- section 2.3: the mathematical description of how persistence is achieved through a Hinge factor is hard to follow. The mathematical formulation as an energy potential term, or a force term, should be reported in the main text.
- section 2.4: I wouldn't express the way thethers are modeled as a "force of inertia". It seems to me that the authors modeled thethers as an "higher viscous drag".
Author Response

(The authors gave the same response as above.)

Reviewer 3 Report
Comments and Suggestions for Authors
This work by Kolbin et al., uses polymer modeling to study how condensin-mediated DNA looping is influenced by chromatin stiffness, condensin strength, and DNA tethering. The simulations reveal that condensin strength dominates loop size when DNA is firmly tethered, chromatin stiffness dominates when untethered, and both contribute to intermediate tethering regimes. This provides insights into how loop patterns and sizes emerge from molecular properties and interactions between chromatin, condensin complexes, and DNA tethering proteins. Overall, it’s a well written manuscript. However, it needs some significant improvements/validations/control experiment before it can be published.
1. The condensin complexes are modeled as simple Hookean springs. A more detailed model accounting for condensin's architecture (e.g. its ring-like shape) could provide insights into how its structure facilitates loop extrusion.
2. The effects of ATP hydrolysis by condensin are not incorporated into the model. Adding an energy term and rules for ATP binding/hydrolysis could reveal mechanistic insights into how condensins translocate along DNA.
3. The model currently focuses only on condensin-mediated looping. Expanding it to include cohesin and other loop extruding factors could shed light on their coordinated effects on chromatin organization.
4. The stiffness of the tethers at chromatin domain boundaries is modeled simply as a drag force. A more detailed representation as proteins or DNA-protein complexes could reveal how tether mechanics regulate domain looping.
5. The model could be expanded to 3D to investigate how chromosome folding in 3D space impacts looping patterns. This could provide insights into heterochromatin clustering and territory formation.
6. More details on the kinetics and thermodynamics of the looping, such as rates of loop formation and breakdown, could provide additional insights into chromatin dynamics.
7. Comparison to high-resolution experimental data, such as Hi-C maps, could help validate and refine the model parameters and assumptions.
8. Analysis of how loop size distributions depend on system size could reveal insights into looping patterns genome-wide vs locally.
9. Incorporate epigenetic features like histone modifications and DNA methylation into the model. These can alter chromatin stiffness and condensin binding affinity, impacting looping.
10. Add stochasticity to condensin binding/unbinding kinetics. This could reveal variability in looping patterns across cell populations.
11. Simulate the effects of DNA replication on looping patterns. This could reveal how loops are disrupted or re-formed during S phase.
12. The model uses a simple bead-spring representation of chromatin. It may be worth exploring more detailed models that incorporate nucleosomes and linker DNA explicitly. This could reveal insights into how loop extrusion and chromatin compaction depend on nucleosome positioning and occupancy.
13. The texts used in the figures are inconsistent. Please make them consistent.
Comments on the Quality of English Language
No major issues detected.
Author Response

(The authors gave the same response as above.)

Round 2
Reviewer 3 Report
Comments and Suggestions for Authors
Some of the more detailed aspects I proposed like condensin architecture, ATP hydrolysis effects, 3D nuclear geometry, could shift the focus away from the key high-level insights you aim to convey in this work about the interplay between chromatin stiffness, condensin strength, and DNA tethering. However, the authors should address the following concerns suggested earlier in this study.
· more details on the kinetics and thermodynamics of looping, such as rates of loop formation and breakdown. This could provide additional insights into chromatin dynamics.
· Analysis of how loop size distributions depend on system size could reveal insights into looping patterns genome-wide vs locally.
· Additionally, the texts used in the figures are still inconsistent. Please make them uniform in size and font.
Author Response
To Reviewer,
Thank you for highlighting important aspects of our work with your comments. Please see the PDF attached for a detailed response to your thoughtful suggestions.

Round 3
Reviewer 3 Report
Comments and Suggestions for Authors
The authors have addressed some of the concerns, and the article can proceed for publication. Please ensure that in the final PDF, the text inside the figures is consistent and clearly visible.